# A warm-white light-emitting diode based on single-component emitter aromatic carbon nitride

Yunhu Wang[1], Kunpeng Wang [1], Fangxu Dai[1], Kai Zhang[1], Haifeng Tang[1], Lei Wang [1,2] ✉ & Jun Xing [1] ✉

Artificial lighting consumes almost one-fifth of global electricity. As an efficient solid-state lighting technology, white light-emitting diodes (WLEDs) have received increasing attention. However, the white luminescence of the traditional WLEDs comes from multi-component emitters, which leads to complex device structure and unstable emitting color. Therefore, developing single-component materials with white-light electroluminescence is of significance for artificial lighting applications. Here, we fabricate single-component white-light electroluminescence devices based on an aromatic carbon nitride material and improve the performance of WLEDs by adjusting the carrier transport. The carbon nitride LEDs emit warm-white light, of which color coordinates and color temperature are (0.44, 0.52) and 3700 K. The optimized LEDs display a very low turn-on voltage of 3.2 V and achieve a milestone in the maximum luminance and external quantum efficiency of 1885 cd m$^{-2}$ and 1.20%. Our findings demonstrate the low-cost carbon nitride materials have promising potential for single-component WLEDs application.

White light-emitting diodes (WLEDs) as efficient solid-state lighting increasingly replace incandescent bulbs and fluorescent tubes to tackle the issue of energy conservation and carbon neutrality[1–4]. The commercial WLEDs are multi-color hybrid devices composed of blue-emissive LEDs chips and yellow-emissive phosphors coating, which bring unstable emission color over time due to the different degradation rates of the emitters and energy losses due to the overlapping absorption. Therefore, single-component materials with broadband electroluminescence (EL) covering the visible spectrum are desirable for the next generation of artificial lighting, which can settle these problems and further simplify the manufacturing process[4]. In contrast, such materials are extremely rare, because electron–hole pairs tend to attain the lowest energy states and result in monochromatic emission[5]. There are only several white light organic LEDs were fabricated based on a single organic emitter, such as 1,3,5-tris(2-(9-ethylcarbazyl-3) ethylene)benzene, 4,40-di(9-(10-pyrenylanthracene)) triphenylamine,

tris(4-(phenylethynyl)phenyl)amine, etc.[6–8]. Recently, various low dimensional metal halide perovskites with self-trapped excitons were demonstrated broadband white light emission, however, there are few reports of EL from these materials, and the peak luminance of the WLEDs is only around 70 cd m$^{-2}$, and no efficiency was recorded[9–15].

Graphitic carbon nitride (g-CN) is a kind of two-dimensional polymeric semiconductor with a simple and low-cost synthesis procedure, low toxicity, and good chemical stability[16–21]. The g-CN materials exhibit a broadband spectrum, which are derived from that radiative transition between the lone pair valence band in the edge N 2p orbitals and the antibonding orbitals conduction band of sp$^2$ C−N bonds[22,23]. Moreover, their emission spectra can be adjusted in the visible light range through structure manipulation, heteroatom doping, copolymerization, etc., highlighting the potential of g-CN as a white light emitter[19,24–26]. Encouraged by these merits, g-CN materials are pursued in lighting applications, however, most works only

[1]Key Laboratory of Eco-chemical Engineering, Ministry of Education, College of Chemistry and Molecular Engineering, Qingdao University of Science & Technology, 266042 Qingdao, China. [2]Shandong Engineering Research Center for Marine Environment Corrosion and Safety Protection, College of Environment and Safety Engineering, Qingdao University of Science & Technology, 266042 Qingdao, China. ✉e-mail: inorchemwl@126.com; xingjun@qust.edu.cn

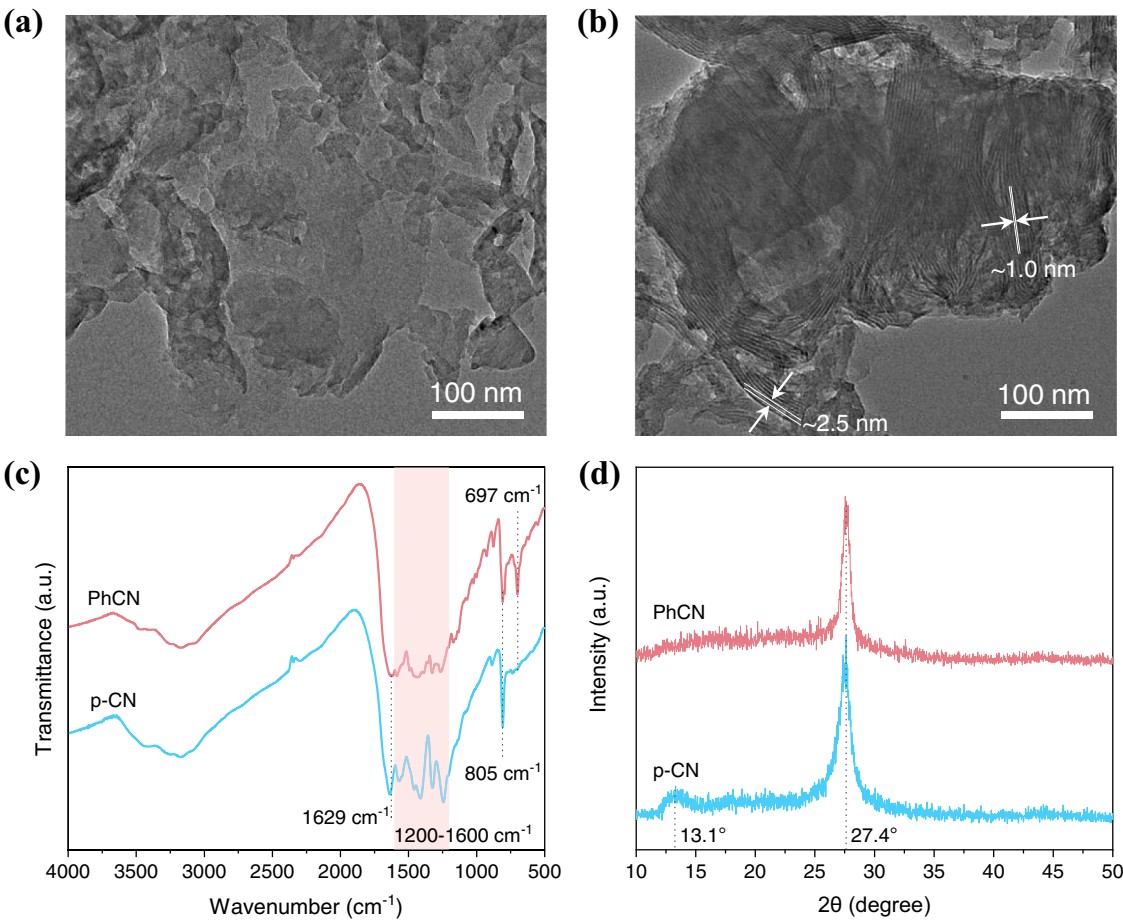

**Fig. 1 | Structures of g-CN materials.** TEM images of **a** p-CN and **b** PhCN. **c** FT-IR and **d** XRD patterns of p-CN and PhCN.

investigate its photoluminescence properties[27–31]. So far, it is still a challenge to fabricate a high-performance EL device based on g-CN materials.

Herein, we first demonstrate an efficient and bright warm-white EL device from aromatic g-CN as a single-component emitter. The phenyl modification can enhances the rigidity of the tri-s-triazine structure, weakens the electron-phonon coupling, thus significantly improve photoluminescence quantum yield (PLQY) from 5 to 40%. The PL spectrum of the phenyl-modified g-CN (PhCN) covers 425–750 nm with wide full width at half maximum (FWHM) of 100 nm. The WLEDs based on the PhCN were fabricated by a vacuum-deposited process, and their performances were improved by optimizing the charge injection. In consequence, these devices exhibit a low turn-on voltage of 3.2 V and reach a milestone in luminance of 1885 cd m$^{-2}$ and external quantum efficiency (EQE) of 1.20% (power efficiency is 2.05 lm W$^{-1}$). The PhCN-based LEDs display warm-white light emission with FWHM of 110 nm, chromaticity coordinates of (0.44, 0.52) and a color temperature of 3700 K.

## Results
### Structure and optical properties of g-CN materials
The pristine g-CN (p-CN) and PhCN samples were synthesized by thermal polymerization using melamine and 2, 4-diamino-6-phenyl-1,3,5-triazine as precursors, respectively. As shown in Fig. 1a, the transmission electron microscopy (TEM) image shows a typical two-dimensional layered structure of p-CN. On the contrary, TEM image of PhCN (Fig. 1b) exhibits a distinct banded structure, which aggregates into layer structure in the end. The width of the PhCN band is approximate 2.5 nm with a band spacing of about 1.0 nm. It can be speculated that the benzene ring of aromatic melamine breaks the

ordered polymerization rule of CN network. The Fourier transform infrared (FT-IR) spectra of p-CN and PhCN exhibit strong vibration bands at 1200–1600 cm$^{-1}$ and 805 cm$^{-1}$ (Fig. 1c), which attribute to the stretching modes of the CN heterocycles and the vibration of the heptazine units, respectively[19,25,26,32,33]. Importantly, compared with p-CN, PhCN showed a bending vibration of the C−H bond in the phenyl group at 697 cm$^{-1}$, indicating that the phenyl groups bonding with the CN network[19,25,32,33]. Meanwhile, due to the conjugation effect of the benzene ring, a wide-stretching vibration band of benzene ring can be observed near 1629 cm$^{-1}$, which also indicates the successful combination of benzene group and CN network[32]. Solid-state $^{13}$C NMR tests were performed to further confirm the introduction of the benzene in PhCN. As shown in Supplementary Fig. 1, both p-CN and PhCN possess characteristic peaks of tri-s-triazine-based g-CN at 156.4 and 164.3 ppm, which were attributed to -CN$_3$ and -CN$_2$(NH$_x$) of tri-s-triazine, respectively[34,35]. An additional characteristic peak belonging to benzene carbon was observed at 110.6 ppm in PhCN, indicating the successful incorporation of the benzene[34]. The X-ray diffraction (XRD) patterns of p-CN and PhCN have the same diffraction peak at 27.4° (Fig. 1d), which can be indexed to the (002) graphite-like plane of the conjugated aromatic system stacking and the corresponding interlayer distance was calculated to be 0.326 nm[17–19,25,27]. The p-CN has one more diffraction peak at 13.1°, resulting from the periodic arrangement of the condensed tri-s-triazine units[17,18,27]. Whereas, the diffraction peak at 13.1° disappears in PhCN. Combined with the above analysis, we can conclude that the presence of benzene will break the polymerization rules of CN network and end the periodic arrangement of tri-s-triazine. This also can be verified by elemental analysis (Supplementary Table 1). The C/N molar ratio is 0.778 for p-CN, closing to the ideal g-CN materials (C/N = 0.75), but rises to 1.105 for PhCN. On the one hand, the

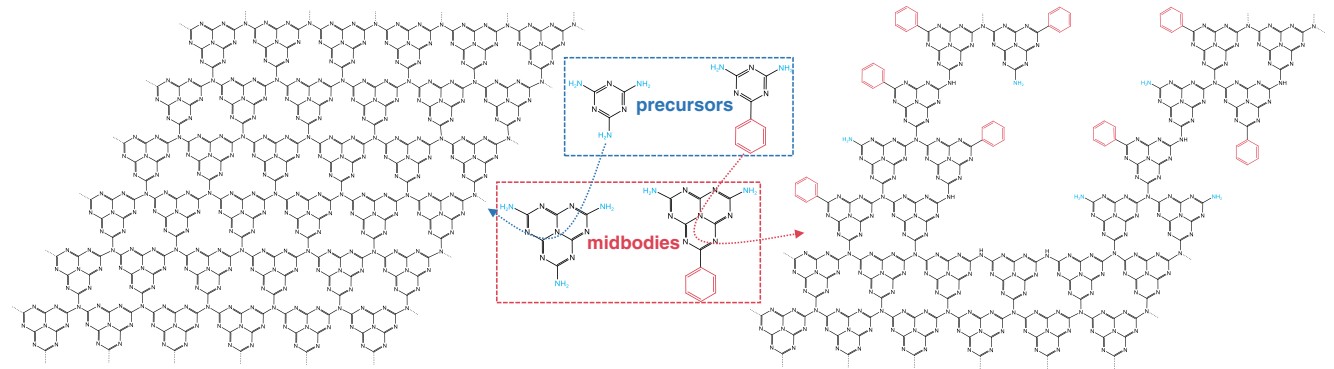

**Fig. 2 | Synthesis process of g-CN materials.** A schematic of p-CN and PhCN prepared via the one-step thermal polymerization process from the precursor.

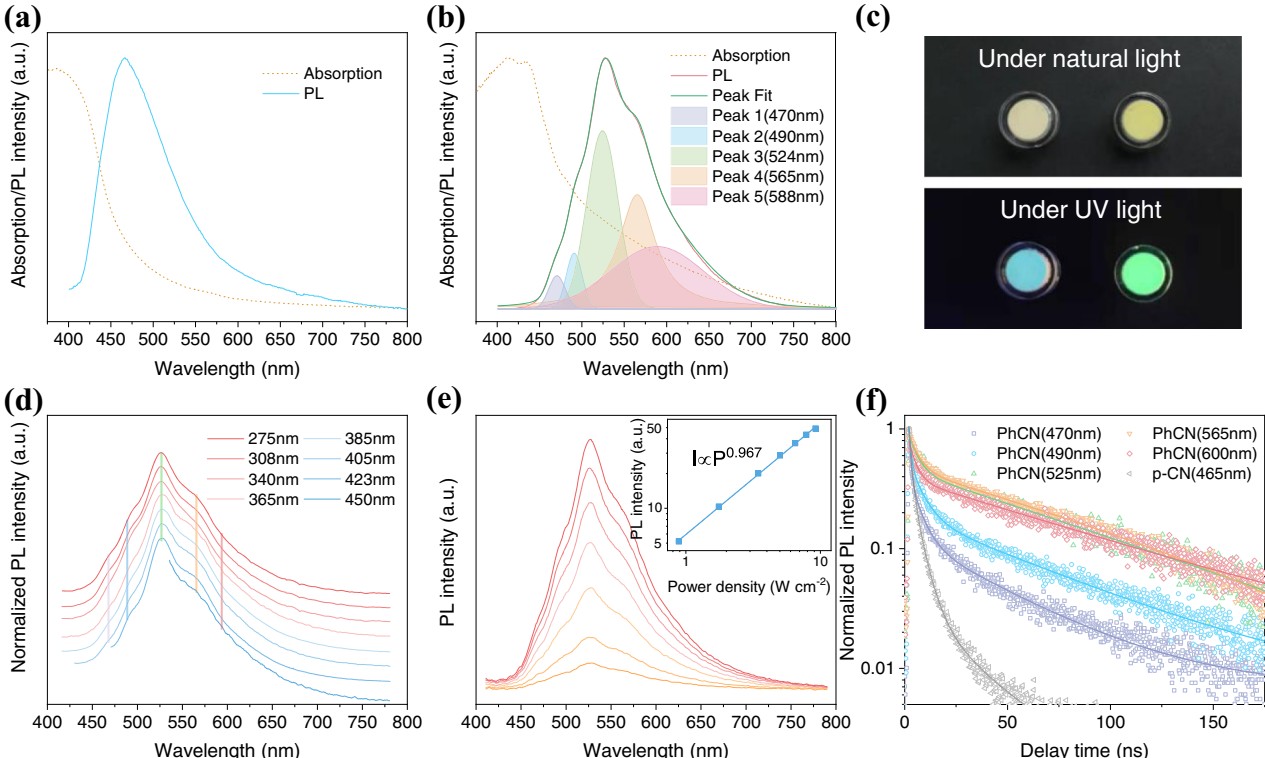

**Fig. 3 | Optical properties of g-CN materials.** UV−Vis absorption and PL spectra of **a** p-CN and **b** PhCN. **c** Photographs of p-CN (left) and PhCN (right) powders under natural light and 365 nm UV light. **d** Normalized PL spectra of PhCN under different excitation light. **e** PL spectra of PhCN under 385 nm excitation light with different intensities, (inset: power-dependent curve of luminescence intensity). **f** PL decay profiles of p-CN and PhCN under a 375 nm picosecond laser.

phenyl groups in the PhCN introduce additional C sources in the CN network and do not participate in the formation of heptazine. On the other hand, the lower polymerization degree jointly determines the C/N molar ratios increment (see Supplementary Note 1 for detailed discussion)[19,26,36]. In addition, we prepared PhCN samples in Ar atmosphere under the same conditions. The same structural and optical properties of the samples synthesized at $N_2$ and Ar atmosphere rules out the N-doping in PhCN during preparation procedure (Supplementary Fig. 2 and Supplementary Table 1). According to the above analysis, the thermal polymerization process and specific structure diagram of p-CN and PhCN are presented in Fig. 2.

Figure 3a, b shows the absorption and PL spectra of the g-CN materials. The p-CN displays a sharp absorption edge located at around 450 nm and emits at 465 nm with an FWHM of 95 nm. In contrast, the absorption spectrum of the PhCN obviously redshift and have a broad absorption tail from 480 to 800 nm. Their bandgaps of

2.75 and 2.28 eV were calculated from the Tauc plot (Supplementary Fig. 3). We preliminarily speculate that the reason for the wide absorption tail of PhCN is that the incorporation of phenyl into the CN network enhances the conjugation degree of the CN network and promotes electron delocalization. Theoretical calculation was carried out to further investigate the absorption in more detail later. The PL spectrum of PhCN includes one major emission peak and several shoulder peaks, with an overall FWHM of 100 nm. It can be fitted into five peaks centered at 470, 490, 524, 565, and 588 nm (Supplementary Table 2), respectively. The first peak at 470 nm can be attributed to the transition between the σ* conduction band and the lone pair valence band, formed through the lone pair electron of the nitride valence band. The second one at 490 nm may take place between the bridge N atoms and the tri-s-triazine, which is indexed to the transition between the σ* conduction band and the π valence band[19,37]. The possibility of the σ*-π transition can be attributed to the introduction of phenyl

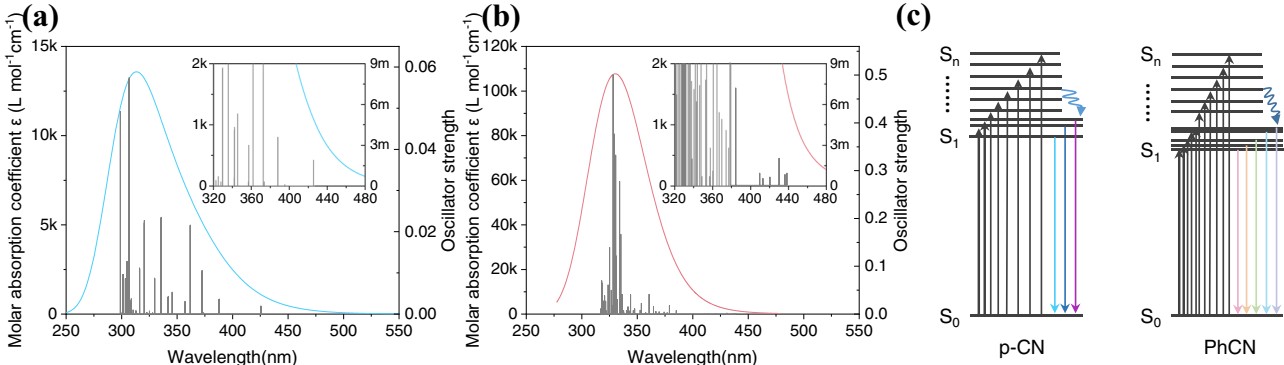

**Fig. 4 | DFT calculations.** The DFT-calculated oscillator strength and molar absorption coefficient of **a** p-CN and **b** PhCN (inset: enlarged detail). **c** Schematics of the excitation and de-excitation processes in p-CN and PhCN. $S_0$, $S_1$, and $S_n$ denote the ground state, the first excited state, and the $n$th excited state, respectively.

groups into the structure of g-CN, which makes a decrease in the energy level of π orbitals along with an enhancement in the rigidity of the tri-s-triazine structure[37]. The third and fourth emissions are derived from the transition from the π* conduction band to the lone pair valence band and π valence band[19,27]. The last one may originate from the phenyl group-induced intermediate state and π valence band. The areas of peaks 3, 4, and 5 account for 92.4% of the whole emission spectra of the sample PhCN. Figure 3c shows the digital photos of p-CN and PhCN powders under natural light and 365 nm excitation light. Under 365 nm UV light, p-CN and PhCN show blue and green color fluorescence emissions, respectively, which are in good agreement with their PL spectra. The PL spectrum of PhCN was measured under different excitation light with a wavelength from 275 to 450 nm, and it remains almost the same (Fig. 3d), suggesting the same excitation and recombination processes occur in the PhCN species with different degrees of polymerization. Figure 3e illustrates the power-dependent PL measurements of PhCN under 385 nm UV light, which shows a linear dependence from 0.9 to 10 W cm$^{-2}$ with a slope of approximately 1, confirming the excitonic luminescence nature of PhCN. Subsequently, time-resolved PL (TR-PL) measurements of p-CN and PhCN were performed under a 375 nm picosecond pulse laser. As shown in Fig. 3f and Supplementary Table 3, the decay profiles can be fitted with a multi-exponential decay function. The p-CN at 465 nm and PhCN at 470 and 490 nm show triple-exponential decay processes. The slowest process can be attributed to the radiative recombination, and the slower and fast features can be attributed to the nonradiative recombination and the electron transfer process between the energy levels, respectively. The radiative recombination lifetimes of p-CN at 465 nm and PhCN at 470 and 490 nm are 23.07, 43.28 and 60.50 ns, respectively. PhCN exhibits a double-exponential decay process at 525, 565, and 600 nm. The long and short lifetimes are attributed to radiative and non-radiative processes, and the corresponding radiative lifetimes are 77.71, 86.37, and 79.07 ns, respectively. As expected, the trend of the radiative recombination lifetimes is the good agreement with the change in PL intensity. Compared with p-CN, the phenyl group in PhCN would promote the electron delocalization of the CN network, reduce the potential barrier of the electron transfer, as well as improves the rigidity of the tri-s-triazine structure, weakens the electron-phonon coupling and fluorescence quenching, thus significantly improving the radiative recombination of electrons–holes in PhCN. The PLQY of the PhCN was measured to be 40%, which is 8 times higher than that of p-CN (5%).

## Physical properties of g-CN materials

The time-dependent density functional theory (TD-DFT) simulations were applied to calculate the excitation energies and oscillator strength of the g-CN materials. As shown in Fig. 4a, b, the oscillator strength of PhCN is significantly higher than that of p-CN, indicating that the phenyl groups make PhCN absorbs more photons. The corresponding molar absorption coefficient ($\varepsilon$) curve can be obtained by Gaussian broadening of the oscillator strength. Accordingly, the molar absorption coefficient of PhCN is higher than that of p-CN. It is worth noting that, PhCN has relative weak oscillator strength at a longer wavelength, indicating that the benzene ring can "activate" the CN network. Therefore, the absorption spectrum of PhCN redshifts and shows an absorption tail at longer wavelength. Furthermore, we studied their valence electron structures by ultraviolet photoelectron spectroscopy (UPS) and obtained their HOMO and LUMO levels, as shown in Supplementary Fig. 4. The HOMO energy level of PhCN almost keeps the same as p-CN, but the LOMO energy level decreases from −3.35 eV for p-CN to −3.80 eV for PhCN. Therefore, the addition of electron-rich group phenyl in the CN network has a great impact on the optical properties of p-CN, reducing bandgap energy, improving PLQY, and prolonging the radiative recombination life. According to the above results, we can picture the excitation and de-excitation processes as shown in Fig. 4c.

To study the physical properties of g-CN materials and provide a theoretical basis for designing LEDs device structure, Mott−Schottky (M-S) tests were conducted, and the electron and hole mobility of the sample were calculated by space charge limited current (SCLC) method, which was widely used for organic semiconductors[38,39]. Figure 5a shows the M-S curves of p-CN and PhCN measured at 1000 Hz. The positive slope of the M-S diagram reveals that both p-CN and PhCN have n-type semiconductor characteristics. Carrier mobility $\mu$ can be calculated by the following equation:

$$J = \frac{9}{8}\varepsilon_0\varepsilon_r\mu\frac{V^2}{d^3} \qquad (1)$$

Where $J$ is the current density, $\varepsilon_0$ is the vacuum permittivity (8.85 × 10$^{-14}$ F cm$^{-1}$), $\varepsilon_r$ is the relative dielectric constant (generally take 3 for organic semiconductors), μ is the carrier mobility, $V$ is the applied bias, and $d$ is the film thickness. As shown in Fig. 5b, c, the electron mobility ($\mu_{e-}$) and hole mobility ($\mu_{h+}$) of PhCN were, respectively, estimated by indium tin oxide (ITO)/PhCN (50 nm)/Ag (40 nm) and ITO/poly(3,4-ethylene dioxythiophene)-poly(styrene sulfonate) (PEDOT:PSS)/PhCN (50 nm)/Ag (40 nm) device structures in the dark. The results show that the $\mu_{e-}$ (1.21 × 10$^{-3}$ cm$^2$ V$^{-1}$ S$^{-1}$) of PhCN is about three times higher than the $\mu_{h+}$ (4.07 × 10$^{-4}$ cm$^2$ V$^{-1}$ S$^{-1}$), which is consistent with the characteristics of n-type semiconductor.

## The g-CN-based LED devices

To demonstrate the potential application of PhCN in LEDs, a series of EL devices were fabricated by using PhCN as a single-component

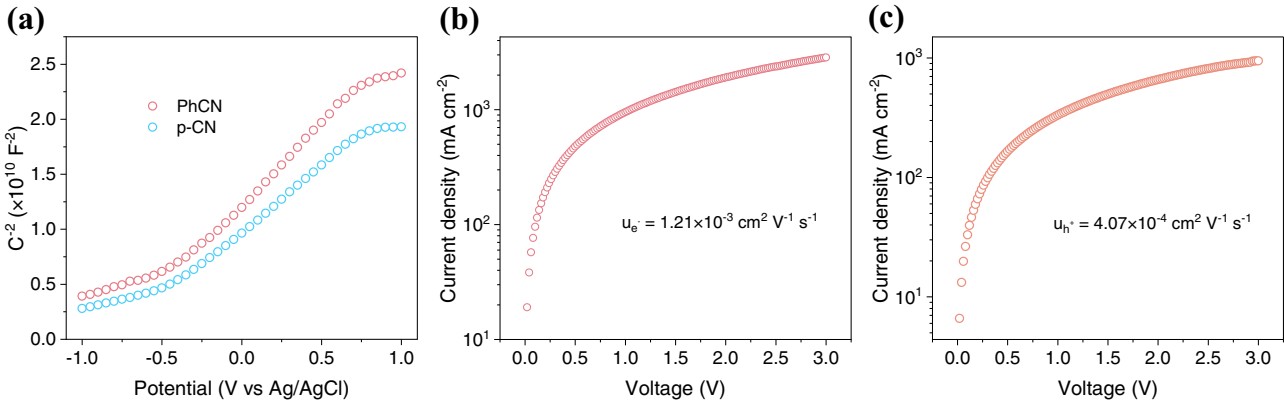

**Fig. 5 | Physical properties of g-CN materials. a** Mott–Schottky plots of p-CN and PhCN at 1000 Hz frequencies. **b** Electron and **c** hole mobility of PhCN were calculated by the SCLC method.

emitting layer. In the first place, we fabricated the LED with a device configuration of the ITO/PEDOT:PSS/Poly(9-vinylcarbazole) (PVK)/PhCN (5 nm)/2,2′,2″-(1,3,5-Benzinetriyl)-tris(1-phenyl-1-H-enzimidazole) (TPBi, 40 nm)/LiF (1 nm)/Al (80 nm) (denoted as device I, Fig. 6a). Atomic force microscopy (AFM) measurement illustrate PhCN films deposited on the surface of PEDOT:PSS/PVK uniformly (Fig. 7a) and the roughness is about 1.47 nm, which meets the requirements of device preparation. Figure 6d shows that the current density of device I increases uniformly with the increase of bias voltage, and the maximum current density is up to 279 mA cm⁻², indicating favorable electron and hole injection properties in the device. Device I has a low turn-on voltage of about 4 V, and reaches the maximum luminance of 915 cd m⁻² at 6.6 V (Fig. 6e). The device shows bright warm-white light emission with the FWHM of 120 nm, CIE chromaticity coordinate of (0.42, 0.51), and corresponding color temperature of 3990 K (Fig. 6i). The efficiency is relatively poor, the maximum EQE and power efficiency are only 0.23% and 0.37 lm W⁻¹, respectively (Fig. 6f). As illustrated in Fig. 6g, the EL spectra show negligible shift with the increase of bias voltage. The main emission peak of the EL spectrum of the device is in the range of 565–573 nm, which indicates that π* to π transition is more likely to occur under the excitation of bias voltage, and also demonstrates that the emission of the device is derived from PhCN. In addition to the above-mentioned, a small emission peak was observed at 450 nm in the EL spectrum, which was speculated to be caused by PVK emission.

Since the electron mobility of PhCN is higher than the hole mobility, we can improve the carrier balance of LEDs by improving the hole injection and reduce the electron injection. Firstly, we added an interlayer Poly[(9,9-dioctylfluorenyl-2,7-diyl)-co-(4,4′-(N-(4-sec-butylphenyl)diphenylamine)] (TFB) in HTL, which not only has an intermediate energy level between PEDOT:PSS and PVK (device II, Fig. 6b), but also has higher hole mobility than PVK[40–42], which can lower the barrier of the holes transport and improve the hole injection ability. AFM test (Fig. 7b) shows that the roughness of PhCN film on this HTL is about 1.63 nm, indicating that the TFB insertion does not affect the deposition of PhCN film. The hole-only devices were prepared with structures of ITO/PEDOT:PSS/PVK/PhCN (5 nm)/ Molybdenum trioxide (MoO₃) (40 nm)/LiF (1 nm)/Al (80 nm) and ITO/PEDOT:PSS/TFB/PVK/PhCN (5 nm)/MoO₃ (40 nm)/LiF (1 nm)/Al (80 nm). The current density–voltage curves show (Supplementary Fig. 5) that the current density of the devices increases significantly after adding TFB. After optimization, the current density of device II (Fig. 6d) increases obviously compared with device I, and the maximum current density reaches 850 mA cm⁻². Device II has a low turn-on voltage of 3.8 V, high luminance of 1552 cd m⁻² at 8.2 V (Fig. 6e), CIE chromaticity coordinates and corresponding color temperature of (0.39,0.48) and 4278 K (Fig. 6i), respectively. Compared with

device I, the performance of device II increased slightly, with the maximum EQE and power efficiency of 0.29% and 0.40 lm W⁻¹, respectively (Fig. 6f). As shown in Supplementary Fig. 6a, the EL spectra of devices I and II were basically the same, the addition of TFB does not change the EL characteristic of PhCN. Meanwhile, a control device using p-CN as the emitting layer was fabricated (denoted as device IIc, Fig. 6b), which shows very poor performance. The device IIc emits blue-purple luminescence with peak luminance, maximum EQE, and power efficiency of 295 cd m⁻², 0.06%, and 0.04 lm W⁻¹, respectively (Fig. 6e, f, i and Supplementary Fig. 6b). AFM image (Supplementary Fig. 7) shows that the p-CN film is uniformly covered with roughness of 1.54 nm, which excludes the effect of film quality on device performance. The EL spectrum of device IIc contains a main emission peak at about 450 nm and a shoulder peak at about 520 nm. To verify the EL origin of device I, II and IIc at 450 nm, we prepared LEDs devices without g-CN materials. Device I w/o g-CN and device II w/o g-CN have EL spectra with emission center of about 430–450 nm (Supplementary Fig. 8). Therefore, we infer that the EL of devices I, II and IIc at about 450 nm comes from PVK emission. In contrast, the EL shoulder peak of device IIc at about 520 nm can be attributed to the p-CN emission, which indicates the poor electroluminescent property of the p-CN materials.

Second, TPBi was replaced by 4,6-Bis(3,5-di(pyridine-3-yl)phenyl) −2-methyl pyrimidine (B3PYMPM) with lower LUMO and smaller electron mobility to reduce the electron injection capacity (device III, Fig. 6c)[43,44], of which the cross-sectional scanning electron microscope (SEM) image is shown in Fig. 7c. Besides, B3PYMPM has a low HOMO energy level, which is benefiting to block holes and reduce the interfacial nonradiative recombination of electrons–holes. Therefore, the current density of device III decreases significantly compared with that of device I and II (Fig. 6d), which can also be verified with electronic-only devices (Supplementary Fig. 9). Due to the low LUMO energy level, the electrons can inject from B3PYMPM to PhCN under lower voltage. Thus, device III exhibits an extremely low turn-on voltage of 3.2 V and reaches a bright warm-white light emission with a maximum luminance of 1885 cd m⁻² at 10.6 V and the maximum EQE and power efficiency are 1.20% and 2.05 lm W⁻¹, respectively (Fig. 6e, f and Supplementary Video 1), which is much better than that of previous reports on g-CN-based LEDs (Table 1). The FWHM of the EL is 110 nm (Fig. 6h), the CIE coordinate is (0.44, 0.52) and the corresponding color temperature is about 3700 K (Fig. 6i). A device III w/o g-CN was fabricated and tested to prove that the EL of device III is derived from PhCN (Supplementary Fig. 10). The EL spectra of device I, II and III exhibit "redshift" compared with the PL of PhCN. We fitted the EL spectra of devices I and II with 5 peaks, while the EL spectra of device III were fitted with 4 peaks (Supplementary Fig. 11). The fitted EL peaks around 490, 525, 560, and 590 nm are well consistent with the PL peaks

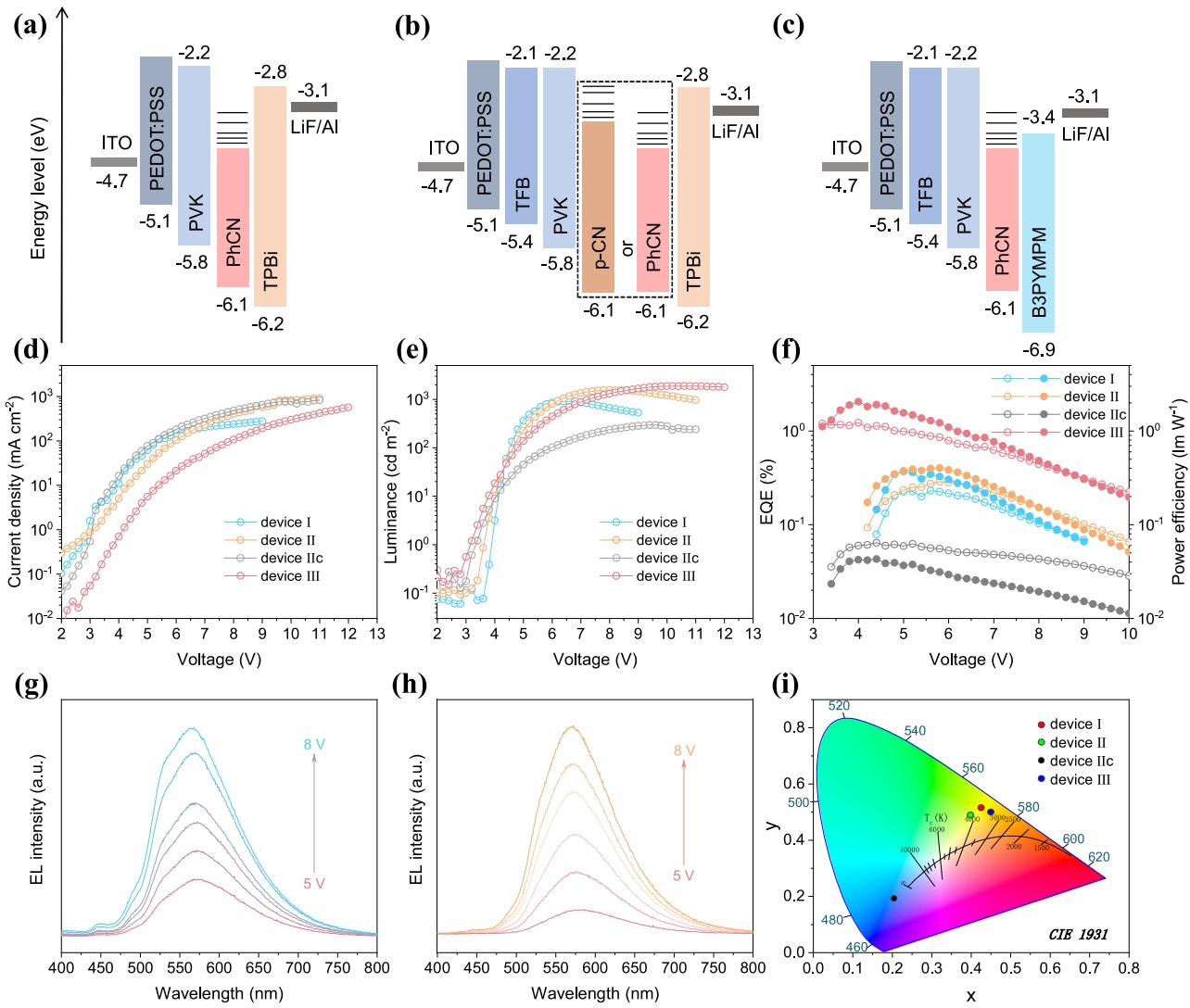

**Fig. 6 | EL performance of the g-CN-based LEDs. a–c** The structures of LEDs devices. **d** Current density-voltage curves, **e** luminance-voltage curves, and **f** EQE-voltage-power efficiency curves (hollow circles represent EQE and solid circles represent power efficiency) of devices I, II, IIc, and III. EL spectra of device I (**g**) and device III (**h**) at various voltages. **i** CIE chromaticity coordinates and corresponding color temperatures for LED devices. The pink, green, black, and blue dots represent devices I, II, IIc, and III, respectively.

position of PhCN (490, 524, 565, and 588 nm), implying the same emitting energy levels in PL and EL processes. It should be noted that the areas of the fitted EL peaks increase as the peak location redshifts, which is inconsistent with their corresponding PL peaks. The EL peak at about 446 nm can be attributed to the emission of PVK. The difference between PL and EL spectra is due to the different recombination dynamics in PL and EL processes. In the PL process, the electrons in the ground state absorb photons and are excited to a high energy level. The excited electrons will relax to lower energy levels gradually; meanwhile, some electrons would recombine during the relaxing process. In the EL process, the electrons transport from ETL to the PhCN, which are preferentially injected into the same and lower energy level of PhCN. The injected electrons will also relax to lower energy levels and recombine meanwhile. Therefore, compared with the PL process, more electrons recombine in lower energy levels in the EL process, causing more emissions at longer wavelengths (denoted as "redshift"). The lower the LOMO of ETL, the more "redshift" of EL (TPBi vs B3PYMPM).

As shown in Supplementary Fig. 12, the device operational lifetime of device III was tested in constant current mode. Unfortunately, the lifetime of the LEDs was only a dozen minutes. It

should be emphasized that the spectra of PhCN-based LEDs are very stable, and there is no obvious color shifting occurring during the operating process. Of course, more efforts are required to in-depth understand and optimize the g-CN materials and their LEDs devices.

## Discussion

In summary, a series of warm-white LEDs were prepared by using PhCN as a single-component emitting layer. By adjusting the hole and electron transport layers, we achieved a champion device with the structure of ITO/PEDOT:PSS/TFB/PVK/PhCN/B3PYMPM/LiF/Al. The optimized LEDs show high performance in turn-on voltage, luminance, and EQE. The devices emit warm-white light with CIE chromaticity coordinate of (0.44, 0.52) and color temperature of 3700 K, which indicate its great potential applications in indoor lighting. The g-CN is becoming one promising luminescent material in the field of EL devices after organic small molecules, polymer molecules and perovskite, etc. Further investigations are certainly required to fully understand and improve the EL properties of the g-CN materials.

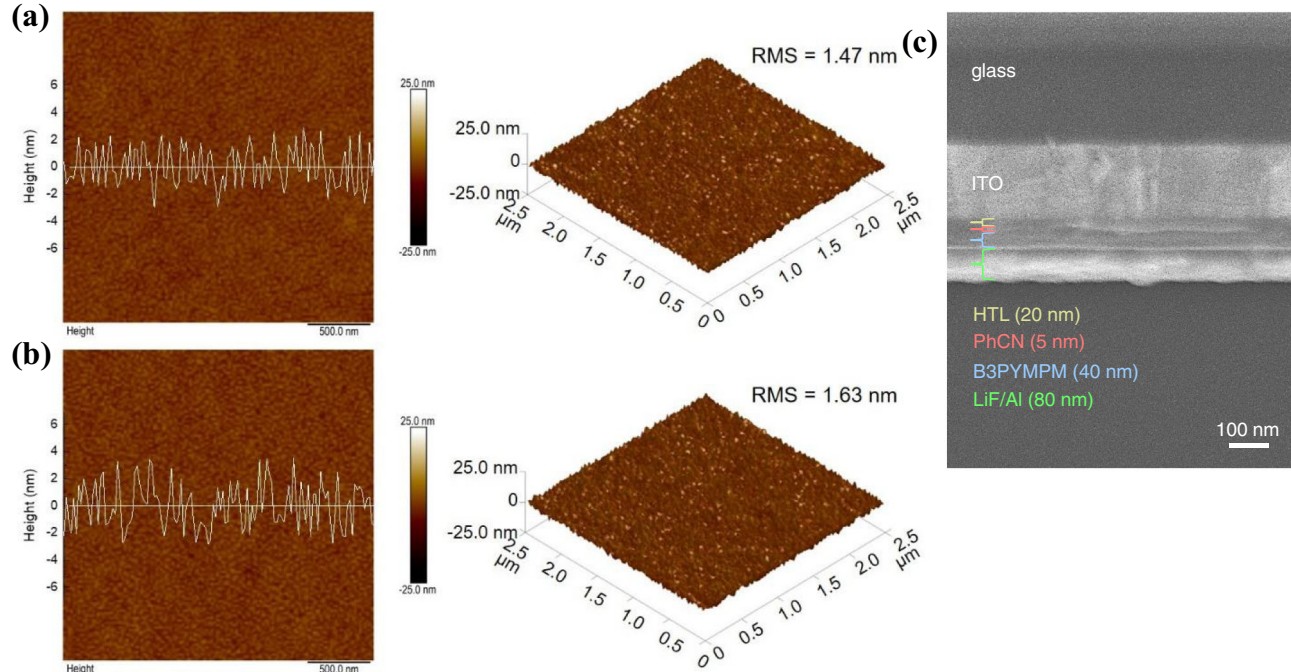

**Fig. 7 | Morphology of PhCN films and device.** AFM images of PhCN films deposited on **a** PEDOT:PSS/PVK and **b** PEDOT:PSS/TFB/PVK surfaces, including height images (left), corresponding line-scan profile images and pseudo-three-dimensional images (right). **c** Cross-sectional SEM image of device III.

### Table 1 | Summary of the performance of g-CN-based LEDs devices

| Precursors | Preparation methods | EL range (nm) | $V_{turn-on}$ @ 1 cd m$^{-2}$ | Max. luminance (cd m$^{-2}$) | Max. EQE (%) | Max. power efficiency (lm W$^{-1}$) | CIE/CCT | Year |
|---|---|---|---|---|---|---|---|---|
| Cyanuric acid/2,4-diamino-6-phenyl-1,3,5-triazine | In situ growth | 600–1000 | ~4.5 V | – | – | – | –/– | 2015[27] |
| Melamine | Spin-coating | 400–600 | 14 V | 11 @ 21 V | – | – | –/– | 2018[28] |
| Melamine | Spin-coating | 400–600 | 6 V | 605 @ 9 V | – | – | –/– | 2019[29] |
| Melamine | In situ growth | 400–800 | – | – | – | – | (0.42, 0.46)/ 3681 K | 2020[30] |
| 2,4-diamino-6-phenyl-1,3,5-triazine | Vacuum deposition | 450–750 | 3.2 V | 1885 @ 10.6 V | 1.20 | 2.05 | (0.44, 0.52)/ 3696 K | This work |

## Methods

### Materials and chemicals

Melamine (99%), 2, 4-diamino-6-phenyl-1,3,5-triazine (98%) and molybdenum trioxide ($MoO_3$, 99.95%) were purchased from Macklin. Chlorobenzene (CB, 99.5%) was purchased from Aladdin. Poly-(9-vinylcarbazole) (PVK), 2,2′,2″-(1,3,5-Benzinetriyl)-tris(1-phenyl-1-H-enzimidazole) (TPBi) and 4,6-Bis(3,5-di(pyridine-3-yl)phenyl)−2-methyl pyrimidine (B3PYMPM) were purchased from Luminescence Technology Corp. Poly[(9,9-dioctylfluorenyl-2,7-diyl)-co-(4,4′-(N-(4-sec-butylphenyl)diphenylamine)] (TFB) was purchased from American Dye Source, Inc. Poly(3,4-ethylene dioxythiophene)-poly(styrene sulfonate) (PEDOT:PSS) model CLEVIOS P VP Al 4083, purchased from Heraeus. All chemicals were not further purified before use.

### Preparation of the p-CN and PhCN

Specifically, melamine (10 g) was put into an alumina crucible with a cover and then transferred to the tubular furnace, which was continuously heated to 550 °C at a heating rate of 5 °C min$^{-1}$, and reacted in a high purity $N_2$ gas (purity: 99.999%) atmosphere with a flow rate of 10 ml min$^{-1}$ for 2 h. The product was ground into powder to obtain p-CN samples. 2, 4-diamino-6-phenyl-1,3,5-triazine (5 g) was put into an alumina crucible with a cover and transferred to a tubular furnace, then heated at 400 °C for 40 min under a high purity $N_2$ gas or Ar gas (purity: 99.999%) atmosphere (10 ml min$^{-1}$) with a heating rate of 2.3 °C min$^{-1}$. The PhCN product was also ground into powder with a yield of about 20%.

### Preparation of g-CN-based LEDs

For a typical fabrication, a 2 cm × 2 cm ITO glass was washed in an ultrasonic bath with deionized water, acetone, and isopropanol for 10 min each repetition and treated with oxygen plasma for 10 min. PEDOT:PSS solution was diluted with ultra-pure water in a ratio of 1:1, followed by spin-coating on ITO glass at 4000 rpm for 60 s, then annealed at 150 °C for 20 min in air. TFB or PVK was dissolved in chlorobenzene (CB) with a concentration of 2.5 mg ml$^{-1}$ for standby. In the nitrogen-filled glovebox, TFB/CB solution was spin-coating at 2000 rpm for 45 s, then annealed at 150 °C for 30 min. After that, CB (70 µL) was dropped onto the substrate coated with TFB film, then spin-coated at 2000 rpm for 45 s, and annealed at 120 °C for 30 s to obtain the thin TFB film. PVK/CB (70 µL) solution was then spin-coating at 4000 rpm for 60 s and annealed at 120 °C for 20 min in the glovebox. Finally, the substrate was transferred into a vacuum deposition chamber, and PhCN or p-CN (5 nm),

TPBi or B3PYMPM (40 nm), LiF (1 nm), and Al (80 nm) were sequentially deposited by thermal evaporation in a vacuum of about $5 \times 10^{-7}$ Torr.

## Characterizations

The g-CN material was dispersed in isopropyl alcohol by ultrasonic for 30 min, then dropped on the ultrathin carbon support film and dried. The surface morphology of g-CN was analyzed by transmission electron microscopy (TEM) using JEM-F200 microscope (JEOL, Japan). Fourier transform infrared (FT-IR) spectra were performed on a Nicolette 6700 FIIR spectrophotometer (Thermo, America). X-ray diffraction (XRD) pattern of the powder was determined by using X'Pert-PRO MPD diffractometer (PANalytical B.V., Holland) at 40 KV and 40 mA Cu Kα obtained under radiation. Solid-state $^{13}$C NMR spectra were measured by AVAN CE NEO 400 NMR spectrometer (Bruker, Switzerland). X-ray photoelectron spectroscopy (XPS) data were collected by using a ESCALAB XI+ (Thermo Fisher, America). The elemental analysis (EA) was performed on the Vario EL cube elemental analyzer (Elementar, Germany). The UV/Vis absorption spectra were obtained on Cary 5000 spectrophotometer (Agilent, America). The PL spectra at different excitation wavelengths and different power densities were measured by using QEPro spectrometer (Ocean Optics, America). Time-resolved PL (TR-PL) spectra were measured by using FLS1000 fluorescence spectrometer (Edinburgh Instruments, UK) and a 375 nm picosecond pulse laser. Ultraviolet photoelectron spectroscopy (UPS) performed the tests with Axis Supra photoelectron spectrometer (SHIMADZU, Kratos, UK). The PLQY measurements were performed on a QEPro spectrometer (Ocean Optics, America) equipped with an integrating sphere and a 385 nm LED as excitation light. Mott–Schottky (M-S) curves were measured in the DC potential range at AC amplitude of 5 mV and frequencies of 1000 Hz under dark conditions. Atomic force microscope (AFM) tests were carried out with Bruker SNL-10 Silicon-tip through Multimode8 atomic force microscope (BRUKER, Germany). Scanning electron microscope (SEM) images were collected with Regulus 8100 (Hitachi, Japan).

## DFT calculation

The first-principles calculations were performed based on DFT, as implemented in the Vienna ab initio simulation package. The calculations were performed based on DFT with b3lyp/6-31 g(d). The excitation energy and vibrator intensity from the ground state to a series of excited states are calculated under the optimized structure of the ground state using TD-DFT. Multiwfn and VMD software packages were used for data processing and rendering[45,46].

## LED device performance test

All devices were tested in a glovebox filled with $N_2$. The Keithley 2450 sourcemeter (Keithley, America) provides a stable current and voltage output for the devices. Then QEPro spectrometer (Ocean Optics, America) connecting with an integrating sphere is used to collect the optical signal of the EL device.

## Data availability

The data that support the findings of this study are available from the corresponding author upon reasonable request.

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

## Acknowledgements

This work was supported by the National Natural Science Foundation of China (21905154), Taishan Scholar Program, Outstanding Youth Foundation of Shandong Province, China (ZR2019JQ14), Major Scientific and Technological Innovation Project (2019JZZY020405), Major Basic Research Program of Natural Science Foundation of Shandong Province under Grant (ZR2020ZD09).

## Author contributions

J.X. conceived the idea and designed the experiments. Y.W. prepared the samples and performed the characterization measurements assisted by L.W. F.D. and K.Z. assisted with the optical measurements. Y.W. analyzed the results assisted by H.T. and L.W. K.W. did the DFT calculations. J.X. and Y.W. wrote the manuscript. All authors read and commented on the manuscript.

## Competing interests

The authors declare no competing interests.
