## [Peer Review File · Nature Communications]

REVIEWER COMMENTS

Reviewer #1 (Remarks to the Author):

In this work, Wang et al. report a white light-emitting diode by employing a carbon nitride material. Although the experimental results and corresponding explanations are reasonable, the EQE is low, only about 1.2%, which cannot be called a high-performance WLED. Furthermore, the manuscript is more like a scientific paper, lacking a deep understanding of device physics/photophysics. The reported PLQY of carbon nitride materials is usually lower than 50%, which may not be suitable for electroluminescent devices, and the EL mechanism of electrical-generated excitons is unclear. Therefore, this work might not be suitable for Nature Communications.

1. In Fig. 2b, what is the reason for the broad absorption tail?
2. In Fig. 2c, the PhCN powder under UV excitation light is green, which is inconsistent with the warm-white EL claimed by the authors. What's the reason?
3. The EL exhibits a significant red shift relative to the PL spectrum, which seems to be unconventional. It is necessary to consider that the electrical-generated excitons are injected into the defect levels. For the EL process, the author needs to elaborate the process in depth.
4. For device optimization, the authors tried to use different hole and electron transport layers, but these strategies lack innovation. Understanding of device physics is also inadequate.
5. In Fig. 4f, the EQE values (device III) seem to be inaccurate around 3.0 V because the luminance is still below 1.0 cd/m² at this stage.
6. The EQE obtained is too low, only around 1.2%. Considering the low PLQY and unclear EL process, this carbon nitride material may not be suitable for the fabrication of electroluminescent devices.
7. The authors claimed that EL at 450 nm is derived from the PVK material. In Figure S4b, however, the EL of the control device (device II c) comes from PVK instead of the p-CN emitting layer, so the device II c has no meaning for comparison.
8. The manuscript lacks critical data, such as SEM of the emitting layer, cross-section SEM or TEM of the device, device operational lifetime, etc.

Reviewer #2 (Remarks to the Author):

In this work, the authors fabricate a warm-white LED based on aromatic carbon nitride material and obtain high performance on efficiency and luminance. The results are well analyzed and discussed.

The single-emitter based white LED is important, but really rare. This is an interesting work, which provides a kind of material showing prospects for white LED applications and will inspire researchers to investigate the electroluminescence properties of carbon nitride materials. Therefore, I would like to recommend its publication after addressing my following concerns.

1. For comparison, it is confusing that the authors fabricated pristine and aromatic carbon nitride at different temperatures. The authors should explain it. Can the authors provide the results of the LED device by using the carbon nitride materials produced at the same temperature?

2. The authors fabricated the carbon nitride by using melamine and phenyl-melamine. Compared with melamine, phenyl-melamine has an asymmetry structure, which would influence its polymerization process. It is better to supplement more characterizations to detect the structure of these carbon nitride materials.

3. The HTL materials in the LED device, TFB and PVK, are also electroluminescence materials. The authors should prove that the electroluminescence of the device is derived from carbon nitride rather than TFB or PVK ?

4. The authors emphasize the low cost of carbon nitride materials in the manuscript. As we know, carbon nitride materials can be synthesized by a simple thermal-polymerization method and using low-cost precursors such as melamine, dicyandiamide, and urea. How much is the precursor phenyl-melamine? How much is the product yield of the aromatic carbon nitride material?

Reviewer #3 (Remarks to the Author):

The authors fabricated single-component white-light electroluminescence devices based on an aromatic carbon nitride material and improved their performances by adjusting carrier transport. This approach has some reference value on development of single-component WLEDs. However, there are still some details that haven't been explained clearly. More research needs to be done before publication. The following questions might be helpful for improvement of this work.

point 1: In line 65 the paper, it says that the p-CN and PhCN were synthesized by melamine and 2, 4-diamino-6-phenyl-1,3,5-triazine two molecules respectively, however in Figure 1(a) you can see that the molecules labelled as "precursor" were in fact not melamine and 2, 4-diamino-6-phenyl-1,3,5-triazine but something more complex. What is the real precursor of p-CN and PhCN? What roles do the molecules in Figure 1(a) actually are? Transformation midbodies or something else. This should be clarified.

point 2: It is widely known that when a material is heated in high temperature under nitrogen atmosphere, the nitrogen atom will have possibility to be doped into the materials. For example, the N-doped graphene nanotubes are synthesized in this way. So, in the preparation process will the

nitrogen dopes into your materials during heating, changes the carbon/nitrogen ratio and finally leads to some change in the properties?

Point 3: In lines 155 to 160, there are some errors in the explanation of the reasons for the inconsistency between the EL and PL spectra. The authors believe that the electrons recombine at the lowest energy level under the electroluminescence condition, while the electrons recombine at the higher excited state energy level under the photoluminescence condition. First, under photoluminescence, electrons will still undergo a rapid non radiative transition to the lowest energy level before recombination. Moreover, according to the author's explanation, it will be inconsistent with the result that the PL spectrum remains unchanged under different excitation wavelengths mentioned in line 104 above (The PL spectrum of PhCN was measured under different excitation light with wavelength from 275 to 450 nm, and it remains almost the same). Use the author's theory to analyze: different excitation wavelengths correspond to different photon energies, then the excited state energy levels of electrons should be different, and the PL spectrum will change accordingly, this is in contradiction with the above results. Generally, this spectral change is caused by the change of molecular dipole in the presence of electric field.

Point 4: In line 31, the paper says that the commercial WLEDs composed of blue-emissive LEDs chips and yellow-emissive phosphors coating that “bring unstable emission color over time”, however, there is no data on device stability or device lifetime for phCN-based devices. Supplementary data should be added to prove improved stability, otherwise it is difficult to demonstrate the potential of this material when both EQE and maximum luminance are much lower than those of commercial WLEDs.

Reviewer #1 (Remarks to the Author):

In this work, Wang et al. report a white light-emitting diode by employing a carbon nitride material. Although the experimental results and corresponding explanations are reasonable, (1) the EQE is low, only about 1.2%, which cannot be called a high-performance WLED. (2) Furthermore, the manuscript is more like a scientific paper, lacking a deep understanding of device physics/photophysics. (3) The reported PLQY of carbon nitride materials is usually lower than 50%, which may not be suitable for electroluminescent devices, and (4) the EL mechanism of electrical-generated excitons is unclear. Therefore, this work might not be suitable for Nature Communications.

Response: We thank the reviewer for his/her valuable efforts in reviewing our manuscript. Following the reviewer's suggestions, we performed new experiments and made extensive revisions to strengthen our manuscript. (1) The single-component materials with broadband white electroluminescence (EL) are extremely rare, and so far their performances are far behind the monochromatic LED. Here, to address the reviewer's concern and avoid subjective assessments in the title, we revised the title as follows: A Warm-White Light-Emitting Diode Based on Single-Component Emitter Aromatic Carbon Nitride. (2) We carried out Mott-Schottky (M-S) and electron- or hole-only device tests to elucidate the physical/photophysical mechanism of the device. (3) Normally, the carbon nitride materials are synthesized through the thermal polymerization method using melamine, urea, or thiourea as the precursor, of which the products exhibit low PLQY of less than 10%. However, the best PLQY of the carbon nitride could be improved to >90% by quantum confinement effect and molecular modification [Adv. Mater. 2017, 30, 1704376; Chem. Commun. 2019, 55, 15065; Nanoscale, 2019, 11, 16553; Appl. Surf. Sci. 2019, 504, 144330]. The carbon nitride materials may be promising for electroluminescent devices in view of the high PLQY, which of course requires more effort in in-depth understanding and optimizing the carbon nitride based LEDs. (4) The EL mechanism was analyzed in more detail in the revised manuscript.

Comment 1. In Fig. 2b, what is the reason for the broad absorption tail?

Response: We thank the reviewer for this comment. According to the Tauc plots of p-CN and PhCN (see Fig. S3 in Supplementary Information), the bandgap of PhCN is significantly reduced compared with that of p-CN (2.28 eV vs 2.75 eV), because the addition of benzene ring to CN network increases the conjugation degree of CN network and promotes electron delocalization.

Accordingly, the DFT calculations (Fig. 4 in the revised manuscript) also show that the oscillator strength and molar absorption coefficient of PhCN are significantly higher than those of p-CN and PhCN also has relative weak oscillator strength at longer wavelength, indicating that the benzene ring can "activate" the CN network. Therefore, the absorption spectrum of PhCN redshifts and shows an absorption tail at longer wavelength.

Comment 2. In Fig. 2c, the PhCN powder under UV excitation light is green, which is inconsistent with the warm-white EL claimed by the authors. What's the reason?

Response: We thank the reviewer for the good question. As shown in Fig. R1-1 (also as shown in Fig. S11 in the Supplementary Information), we fitted the EL spectra of device I and II with 5 peaks, while the EL spectra of device III were fitted with 4 peaks. The fitted EL peaks around 490, 525, 560 and 590 nm are well consistent with the PL peaks position of PhCN (490, 524, 565 and 588 nm), implying the same emitting energy levels in PL and EL processes. It should be noted that the areas of the fitted EL peaks increase as the peak location redshifts, which is inconsistent with their corresponding fitted PL peaks. The EL peak at about 446 nm can be attributed to the emission of charge transport layer material (see revised manuscript for the detailed discussion). The difference between PL and EL spectra is due to their different recombination dynamics in PL and EL processes.

As shown in Fig. R1-2, in the PL process, the electrons in the ground state absorb photons and are excited to high energy level. The excited electrons will relax to lower energy levels gradually; meanwhile, some electrons would recombine during the relaxing process. In the EL process, the electrons transport from ETL to the PhCN, which are preferentially injected into the same and lower energy level of PhCN. The injected electrons will also relax to lower energy levels and recombine meanwhile. Therefore, compared with the PL process, more electrons recombine in lower energy levels in the EL process, causing more emissions at longer wavelengths (denoted as "redshift"). The lower the LOMO of ETL, the more "redshift" of EL (TPBi vs B3PYMPM).

We extended the above analysis in the revised manuscript (page 12).

Fig. R1-1 (a) PL spectrum of PhCN and EL spectra of devices I (b), II (c) and III (d).

Fig. R1-2 Schematic diagram of electron-hole recombination dynamics in the PL and EL processes.

Comment 3. The EL exhibits a significant red shift relative to the PL spectrum, which seems to be unconventional. It is necessary to consider that the electrical-generated excitons are injected into the defect levels. For the EL process, the author needs to elaborate the process in depth.

Response: We thank the reviewers for their constructive comments. In the manuscript, we conducted the power-dependent PL measurement on PhCN powder (shown in Fig. 3e in the revised manuscript), which shows a linear relationship from 1 to 10 W cm^{-2} with a slope of approximately 1, indicating the excitonic emission of the sample, not the defect luminescence. Considering the defects may be introduced in the PhCN film during the preparation process, we further performed the power-dependent PL on the PhCN film synthesized via thermal evaporation method, which displays the same phenomenon as the PhCN powder (Fig. R1-3). Therefore, we can exclude the luminescent defect in the sample and the EL "redshift" is not induced by the defect levels. In addition, we have analyzed the EL "redshift" in more detail in response 2.

Fig. R1-3 The power-dependent PL spectra of PhCN film under 385 nm UV light (inset: power-dependent PL intensity).

Comment 4. For device optimization, the authors tried to use different hole and electron transport layers, but these strategies lack innovation. Understanding of device physics is also inadequate.

Response: We thank the reviewer for making this comment. The device optimization by designing charge transport layers is indeed well-studied. In this work, considering the unclear optoelectrical properties of PhCN, on the one hand, we improved the performance of the LEDs by optimize the hole and electron transport layer, on the other hand, we want to show the different dynamics of the electron-hole recombination in the devices. Following the reviewer's suggestion, we conducted Mott-Schottky (M-S) tests and electron-only or hole-only devices measurements to understand the device physical. Fig. R1-4a (also shown in Fig. 5a in the revised manuscript) shows the M-S curves of p-CN and PhCN at 1000 Hz. The positive slope reveals that both of them have the characteristics

of n-type semiconductors. Furthermore, the carrier mobility of PhCN was estimated by the space charge limited current (SCLC) method, which was widely used for organic semiconductors [Appl. Phys. Lett. 1999, 74, 1132-1134; Appl. Phys. Lett. 2007, 90, 203512]. The carrier mobility was calculated by the following equation:

$$J = \frac{9}{8} \varepsilon_0 \varepsilon_r \mu \frac{V^2}{d^3} \quad (1)$$

Where J is the current density, ε_0 is the vacuum permittivity (8.85×10^{-14} F cm⁻¹), ε_r is the relative dielectric constant (generally take 3 for organic semiconductors), μ is the carrier mobility, V is the applied bias, and d is the film thickness. In the dark environment, the electron and hole mobility of PhCN were estimated with the device structures of ITO/PhCN (50 nm)/Ag (40 nm) and ITO/PEDOT:PSS/PhCN (50 nm)/Ag (40 nm), respectively. As shown in Fig. R1-4b and c (also shown in Fig. 5b and c in the revised manuscript), the electron and hole mobility of PhCN are about 1.21×10^{-3} and 4.07×10^{-4} cm² V⁻¹ s⁻¹, which is consistent with the characteristics of n-type semiconductor.

Thus, we need to improve the hole transport in HTL and reduce the electron transport in ETL to balance the charge injection in LED device. On the one hand, TFB was added in the interlayer between PEDOT:PSS and PVK to improve the hole injection ability, which can be demonstrated by the hole-only device in Fig. R1-5a (also shown in Fig. S5 in the Supplementary Information). On the other hand, TPBi was replaced by B3PYMPM with lower LUMO and smaller electron mobility to reduce the electron injection capacity significantly, which can be illustrated by the electron-only device in Fig. R1-5b (also shown in Fig. S9 in the Supplementary Information). Finally, we obtained PhCN based LEDs with improved charge balance and enhanced performance.

We have added these results and discussions in the revised manuscript (page 8, 9, 10, 11).

Fig. R1-4 (a) Mott-Schottky plots of p-CN and PhCN at 1000 Hz frequencies. (b) Electron and (c) hole mobility of PhCN estimated by the SCLC method.

Fig. R1-5 Current density-voltage curves of (a) hole-only devices (HOD) and (b) electron-only devices (EOD). HOD structure: ITO/PEDOT:PSS/with or without TFB/PVK/PhCN (5 nm)/MoO₃ (40 nm)/LiF (1 nm)/Al (80 nm), EOD-TPBi structure: ITO/TPBi (20 nm)/PhCN (5 nm)/TPBi (20 nm)/LiF (1 nm)/Al (80 nm), EOD-B3PYMPM structure: ITO/B3PYMPM (20 nm)/PhCN (5 nm)/B3PYMPM (20 nm)/LiF (1 nm)/Al (80 nm).

Comment 5. In Fig. 4f, the EQE values (device III) seem to be inaccurate around 3.0 V because the luminance is still below 1.0 cd/m² at this stage.

Response: We thank the reviewer for this comment. The luminance of device III at 3.0 V is about 0.6 cd m⁻². Considering the accuracy of the data, we have removed this data point in the revised manuscript.

Comment 6. The EQE obtained is too low, only around 1.2%. Considering the low PLQY and unclear EL process, this carbon nitride material may not be suitable for the fabrication of electroluminescent devices.

Response: We appreciate the reviewer for this comment. Carbon nitride materials were generally studied as photocatalysts. The carbon nitride materials with low PLQY are beneficial to the separation of the photo-generated electron-hole and improvement of the photocatalytic reaction. Most of the carbon nitride materials were synthesized through the thermal polymerization method using melamine, urea, or thiourea as the precursor, of which the products exhibit low PLQY of less than 10%. As luminescent materials, carbon nitride was neglected and less investigated. However, the PLQY of the carbon nitride could be improved to >90% by the quantum confinement effect, molecular modifications et al. [Adv. Mater. 2017, 30, 1704376; Chem. Commun. 2019, 55, 15065;

Nanoscale, 2019, 11, 16553; Appl. Surf. Sci. 2019, 504, 144330]. The carbon nitride materials may have application prospects in electroluminescent devices in view of the high PLQY, which of course requires more effort in in-depth understanding the mechanism and optimizing the properties. We further analyzed the EL mechanism in more detail in response 2. Besides, we would like to emphasize the innovation of our work, (1) a very cheap material PhCN was applied to prepare WLEDs for the first time, of which the single-component white light emission is very scarce; (2) The performance including EQE and luminance of the PhCN-based LEDs has been greatly improved compared with previous reported CN-based LEDs, showing the promising prospect of LEDs based on carbon nitride materials.

Comment 7. The authors claimed that EL at 450 nm is derived from the PVK material. In Figure S4b, however, the EL of the control device (device IIc) comes from PVK instead of the p-CN emitting layer, so the device IIc has no meaning for comparison.

Response: We thank the reviewer for this comment. To make clear the EL origin, we prepared LED devices without CN emitting layer. As shown in Fig. R1-6 (also shown in Fig. S8 in the revised manuscript), devices I and II w/o CN layer have EL spectra with emission center of about 430-450 nm. Therefore, we infer that the EL of devices I, II and IIc at about 450 nm comes from PVK. In contrast, device IIc shows an EL shoulder peak at about 520 nm, which can be attributed to the p-CN emission. This result demonstrates the poor electroluminescent property of the p-CN materials and inversely highlights the advancement of PhCN materials. We have added this discussion in the revised manuscript (page 11).

Fig. R1-6 EL spectra of devices I and II without g-CN materials. Device I w/o g-CN:

ITO/PEDOT:PSS/PVK/TPBi (40 nm)/LiF (1 nm)/Al (80 nm), device II w/o g-CN:
ITO/PEDOT:PSS/TFB/PVK/TPBi (40 nm)/LiF (1 nm)/Al (80 nm).

Comment 8. The manuscript lacks critical data, such as SEM of the emitting layer, cross-section SEM or TEM of the device, device operational lifetime, etc.

Response: We appreciate the reviewer for the good comments. Following the reviewer's suggestions, AFM and SEM characterizations were carried out and presented in Fig. R1-7 (also shown in Fig. 7 in the revised manuscript and Fig. S7 in the Supplementary Information). AFM tests show that PhCN and p-CN are uniformly deposited on substrate without pinhole characteristics, and the roughness is about 1.5 nm. Cross-sectional SEM of the device III illustrates the structure of the device and the thickness of various layers.

As shown in Fig. R1-8 (also shown in Fig. S12 in the Supplementary Information), we detected the device operational lifetime and spectral stability of the PhCN-based LEDs with different device structures. Unfortunately, we obtained a short lifetime of the device, tens of minutes. It should be emphasized that the spectra of PhCN-based LEDs are very stable, and there is no obvious color shifting occurring during the operating process. The g-CN materials were generally studied as photocatalysts. As luminescent materials, g-CN was neglected and rarely investigated. However, the g-CN materials deserve more attention due to their high PLQY, ultra-low cost, eco-friendly nature and structural diversity. Of course, more efforts are required to in-depth understand and optimize the g-CN materials and their LEDs devices.

We have added these results and discussions in the revised manuscript (page 9, 10, 11, 12).

Fig. R1-7 AFM images of PhCN films deposited on (a) PEDOT:PSS/PVK, (b) PEDOT:PSS/TFB/PVK surfaces and (c) p-CN films deposited on PEDOT:PSS/TFB/PVK surface. (d) Cross-sectional SEM image of device III.

Fig. R1-8 (a) Device operational lifetime measured in constant current mode. Spectral stability of (b) device III and (c) device VI. Device IV structure: PEDOT:PSS/TFB/PVK (3.5 mg ml^{-1})/PhCN (5 nm)/B3PYMPM (40 nm)/LiF (1 nm)/Al (80 nm); device V structure: PEDOT:PSS/TFB/PVK (5 mg ml^{-1})/PhCN (5 nm)/1,3,5-tri(p-pyrid-3-yl-phenyl)benzene (TpPyPB, 40 nm)/LiF (1 nm)/Al (80 nm);

device VI structure: PEDOT:PSS/TFB/PVK (8 mg ml⁻¹)/PhCN (5 nm)/TpPyPB (40 nm)/LiF (1 nm)/Al (80 nm).

Reviewer #2 (Remarks to the Author):

In this work, the authors fabricate a warm-white LED based on aromatic carbon nitride material and obtain high performance on efficiency and luminance. The results are well analyzed and discussed. The single-emitter based white LED is important, but really rare. This is an interesting work, which provides a kind of material showing prospects for white LED applications and will inspire researchers to investigate the electroluminescence properties of carbon nitride materials. Therefore, I would like to recommend its publication after addressing my following concerns.

Response: We thank the reviewer for the positive comments. In response to the reviewer's constructive comments, we have performed new experiments and made an extensive revision to strengthen the manuscript.

Comment 1. For comparison, it is confusing that the authors fabricated pristine and aromatic carbon nitride at different temperatures. The authors should explain it. Can the authors provide the results of the LED device by using the carbon nitride materials produced at the same temperature?

Response: We appreciate the reviewer for making this good comment. The control sample p-CN was prepared at 550 °C mainly due to the following reasons: as control sample, p-CN should have similar crystal structure to PhCN prepared at 400°C. According to previous research [Sci. Rep. 2013, 3, 1943; Nanoscale 2015, 7, 12343], melamine will polymerize to melem at 400°C (p-CN₄₀₀, Fig. R2-1, see Ref. Nanoscale 2015, 7, 12343 for XRD of melem), layered g-CN with low crystalline at 500°C and layered g-CN with good crystalline above 550°C. Therefore, to obtain the p-CN materials with similar layered structure, we prepared the p-CN using melamine at 550°C.

Following the reviewer's suggestion, we prepared the LED based on the sample p-CN₄₀₀. As shown in Fig. R2-2, the PL center of p-CN₄₀₀ is about 430 nm. The p-CN₄₀₀ based LED exhibits poor performance with turn-on voltage of 3.4 V, maximum luminance of 230 cd m⁻², and maximum EQE of 0.19%. The EL spectra are in good agreement with the EL spectrum of device II w/o g-CN. Therefore, the EL of p-CN₄₀₀ based LED may originate from PVK emission and the sample p-CN₄₀₀ may be difficult to achieve EL emission.

Fig. R2-1 XRD patterns of p-CN_x and PhCN_x, x is the thermal polymerization temperature.

Fig. R2-2 (a) PL spectrum of p-CN₄₀₀. (b) Luminance-current density-voltage curve, (c) EQE-voltage curve and (d) EL spectra of p-CN₄₀₀ based LED (solid lines) and device II w/o g-CN (dashed line). Device structure: ITO/PEDOT:PSS/TFB/PVK/p-CN₄₀₀ (5 nm)/TPBi (40 nm)/LiF (1 nm)/Al (80 nm).

Comment 2. The authors fabricated the carbon nitride by using melamine and phenyl-melamine.

Compared with melamine, phenyl-melamine has an asymmetry structure, which would influence its polymerization process. It is better to supplement more characterizations to detect the structure of these carbon nitride materials.

Response: We thank the reviewer for the constructive comment. Following the reviewer's suggestion, we characterized these g-CN materials by using TEM. As shown in Fig. R2-3 (also shown in Fig. 1a and b in the revised manuscript), p-CN shows a typical two-dimensional layered structure. On the contrary, PhCN exhibits a distinct banded structure, which aggregates into layer structure in the end. The width of the PhCN band is approximate 2.5 nm with a band spacing of about 1.0 nm. It can be speculated that the benzene ring of aromatic melamine breaks the ordered polymerization rule of CN network.

To further confirm the presence of the benzene ring in PhCN, we have carried out solid-state ^{13}C NMR tests on these samples. As shown in Fig. R2-4 (also shown in Fig. S1 in the Supplementary Information), both p-CN and PhCN have two characteristic peaks at 156.4 and 164.3 ppm, which are attributed to $-\text{CN}_3$ and $-\text{CN}_2(\text{NH}_x)$ of tri-s-triazine, respectively. An additional characteristic peak belonging to benzene ring carbon is observed at 110.6 ppm in PhCN, indicating the successful incorporation of the benzene ring. According to the above analysis, the thermal polymerization process and specific structure diagram of p-CN and PhCN are presented in Fig. R2-5 (also shown in Fig. 2 in the revised manuscript).

We have added these data and discussions in the revised manuscript (page 4, 5, 6).

Fig. R2-3 TEM images (a) p-CN and (b) PhCN.

Fig. R2-4 Solid-state ^{13}C NMR spectra of p-CN and PhCN powders.

Fig. R2-5 A schematic of p-CN and PhCN prepared via the one-step thermal polymerization method.

Comment 3. The HTL materials in the LED device, TFB and PVK, are also electroluminescence materials. The authors should prove that the electroluminescence of the device is derived from carbon nitride rather than TFB or PVK?

Response: We thank the reviewer for this suggestion. We prepared a series of LEDs without emitting layer (g-CN material). As shown in Fig. R2-6 (also shown in Fig. S8 and Fig. S10 in the Supplementary Information), the EL centers of device I w/o g-CN, device II w/o g-CN and device III w/o g-CN are about 430, 450 and 500 nm, respectively, which are significantly different from those of devices I, II and III, indicating the EL of the g-CN based LEDs originate from emitting layers (g-CN) rather than the charge transport layers (TFB or PVK). We have added this discussion in the revised manuscript (page 11, 12).

Fig. R2-6. (a) EL spectra of device I and device II with or without g-CN materials. Device I w/o g-CN device structure: ITO/PEDOT:PSS/PVK/TPBi (40 nm)/LiF (1 nm)/Al (80 nm), device II w/o g-CN structure: ITO/PEDOT:PSS/TFB/PVK/TPBi (40 nm)/LiF (1 nm)/Al (80 nm). (b) EL spectra of device III with or without g-CN materials. Device III w/o g-CN structure: ITO/PEDOT:PSS/PVK/B3PYMPM (40 nm)/LiF (1 nm)/Al (80 nm).

Comment 4. The authors emphasize the low cost of carbon nitride materials in the manuscript. As we know, carbon nitride materials can be synthesized by a simple thermal-polymerization method and using low-cost precursors such as melamine, dicyandiamide, and urea. How much is the precursor phenyl-melamine? How much is the product yield of the aromatic carbon nitride material?

Response: We appreciate the reviewer for making this comment. The precursor phenyl melamine was purchased from Macklin Company (<http://www.macklin.cn/products/D822864>). The price of the precursor is 0.04 \$/g, and the yield of the product PhCN is about 20%. Therefore, the cost of the product PhCN including precursor cost and electric power consumption is about 0.3 \$/g, which is much lower than most of the inorganic and organic emitters. We have added the data in the Methods in the revised manuscript.

Reviewer #3 (Remarks to the Author):

The authors fabricated single-component white-light electroluminescence devices based on an aromatic carbon nitride material and improved their performances by adjusting carrier transport. This approach has some reference value on development of single-component WLEDs. However, there are still some details that haven't been explained clearly. More research needs to be done before publication. The following questions might be helpful for improvement of this work.

Response: We thank the reviewer very much for the positive comments. We have responded to detailed comments as follows and made extensive revisions to strengthen this manuscript.

Comment 1: In line 65 the paper, it says that the p-CN and PhCN were synthesized by melamine and 2, 4-diamino-6-phenyl-1,3,5-triazine two molecules respectively, however in Figure 1(a) you can see that the molecules labelled as "precursor" were in fact not melamine and 2, 4-diamino-6-phenyl-1,3,5-triazine but something more complex. What is the real precursor of p-CN and PhCN? What roles do the molecules in Figure 1(a) actually are? Transformation midbodies or something else. This should be clarified.

Response: We thank the reviewer for pointing out this problem. We are sorry to describe this point unclearly. In Fig. 1a, the molecules are midbodies. As the second reviewer's suggestion, we further performed TEM characterization to illustrate the detailed structure of the samples p-CN and PhCN. As shown in Fig. R3-1 (also shown in Fig. 1a and b in the revised manuscript), p-CN shows a typical two-dimensional layered structure. On the contrary, PhCN exhibits a distinct banded structure, which aggregates into layer structure in the end. The width of the PhCN band is approximate 2.5 nm with a band spacing of about 1.0 nm. It can be speculated that the benzene ring of aromatic melamine breaks the ordered polymerization rule of CN network. Combined with the above analysis and NMR, FT-IR, XRD results, we illustrated the thermal polymerization process of p-CN and PhCN in Fig. R3-2 (also shown in Fig. 2 in the revised manuscript), which presents the chemical structure of the precursors, midbodies and products.

Fig. R3-1 TEM images (a) p-CN and (b) PhCN.

Fig. R3-2 A schematic of p-CN and PhCN prepared via the one-step thermal polymerization method.

Comment 2: It is widely known that when a material is heated in high temperature under nitrogen atmosphere, the nitrogen atom will has possibility to be doped into the materials. For example, the N-doped graphene nanotubes are synthesized in this way. So, in the preparation process will the nitrogen dopes into your materials during heating, changes the carbon/nitrogen ratio and finally leads to some change in the properties?

Response: We thank the reviewer for the good comment. In response to the reviewer's comment, we prepared the control sample PhCN_{Ar} in Ar gas atmosphere (see Methods section in the revised manuscript) to explore whether nitrogen was doped into the product during the preparation process. Under the same testing conditions, we performed elemental analysis tests on PhCN_{N₂} and PhCN_{Ar}. As shown in Table R3-1 (also shown in Table S1 in Supplementary Information), the results show that there was a negligible difference in C, N, and H elements content. Furthermore, As shown in Fig. R3-3 (also shown in Fig. S2a in Supplementary Information), X-ray photoelectron spectroscopy (XPS) scanning was performed on both samples. The N 1s spectra of both samples are

nearly the same and contain four peaks at 398.6, 399.8, 400.9, and 404.5 eV, which are corresponding to sp^2 -bonded N in triazine rings (C=N-C), tertiary nitrogen bonded to carbon atoms (N-C₃), amino groups (C-N-H) and charging effects or positive charges localization in the heterocycles (π -excitations), respectively. We also investigated the optical properties of the samples. As shown in Fig. R3-4 below (also shown in Fig. S2b in Supplementary Information), there is no difference in PL spectra between PhCN_{Ar} and PhCN_{N₂}, and no significant variation in PLQY (38% vs 40%). Therefore, combined with the elemental analysis, XPS and optical studies, it can be concluded that N₂ won't be doped into the carbon nitride material during heating and lead to changes in the properties. We have added the results and discussion in revised manuscript (page 6).

Table R3-1 Elemental analysis results of PhCN powders synthesized at N₂ and Ar atmosphere.

Sample	C(wt.%)	N(wt.%)	H(wt.%)	C/N(atom%)
PhCN _{N₂}	45.57	48.08	2.507	1.105
PhCN _{Ar}	45.51	48.29	2.516	1.100

Fig. R3-3 XPS N 1s spectra of PhCN powders synthesized at N₂ and Ar atmosphere.

Fig. R3-4 PL spectra of PhCN powders synthesized at N₂ and Ar atmosphere.

Comment 3: In lines 155 to 160, there are some errors in the explanation of the reasons for the inconsistency between the EL and PL spectra. The authors believe that the electrons recombine at the lowest energy level under the electroluminescence condition, while the electrons recombine at the higher excited state energy level under the photoluminescence condition. First, under photoluminescence, electrons will still undergo a rapid non radiative transition to the lowest energy level before recombination. Moreover, according to the author's explanation, it will be inconsistent with the result that the PL spectrum remains unchanged under different excitation wavelengths mentioned in line 104 above (The PL spectrum of PhCN was measured under different excitation light with wavelength from 275 to 450 nm, and it remains almost the same). Use the author's theory to analyze: different excitation wavelengths correspond to different photon energies, then the excited state energy levels of electrons should be different, and the PL spectrum will change accordingly, this is in contradiction with the above results. Generally, this spectral change is caused by the change of molecular dipole in the presence of electric field.

Response: We thank the reviewer for the constructive comment. Generally, as the reviewer mentioned, the electrons will undergo a rapid transition to the lowest energy level, and recombine and emit with narrow PL spectra. The PL spectrum of PhCN is inhomogeneous broadening and contains several shoulder peaks, which certainly originate from different emitting energy levels. This may be due to the complicated structure of the materials.

As shown in Fig. R3-5 (also as shown in Fig. S11 in Supplementary Information), we fitted the EL spectra of device I and II with 5 peaks, while the EL spectra of device III were fitted with 4

peaks. The fitted EL peaks around 490, 525, 560 and 590 nm are well consistent with the PL peaks position of PhCN (490, 524, 565 and 588 nm), implying the same emitting energy levels in PL and EL processes. It should be noted that the areas of the fitted EL peaks increase as the peak location redshifts, which is inconsistent with their corresponding PL peaks. The EL peak at about 446 nm can be attributed to the emission of charge transport layer material (see revised manuscript page 11-12 for the detailed discussion). The difference between PL and EL spectra is due to the different recombination dynamics in PL and EL processes.

As shown in R3-6, in the PL process, the electrons in the ground state absorb photons and are excited to high energy level. The excited electrons will relax to lower energy levels gradually; meanwhile, some electrons would recombine during the relaxing process. In the EL process, the electrons transport from ETL to the PhCN, which are preferentially injected into the same and lower energy level of PhCN. The injected electrons will also relax to lower energy levels and recombine meanwhile. Therefore, compared with the PL process, more electrons recombine in lower energy levels in the EL process, causing more emissions at longer wavelengths (denoted as "redshift"). The lower the LOMO of ETL, the more "redshift" of EL (TPBi vs B3PYMPM).

The dominant emitting energy levels of PhCN are corresponding to five PL peaks location (470, 490, 525, 560 and 590 nm). PL spectra of PhCN were measured under different excitation light with wavelengths from 275 to 450 nm, which are all higher than the major emissions emitting energy levels. Under excitation, the electrons will relax to lower energy levels and recombine in the major emitting energy levels principally. Thus, it is reasonable to observe unchanged PL spectra under different excitation from 275 to 450 nm.

PhCN is graphite-like layered material. During thermal evaporation, the PhCN sheets are prone to lie on the substrate. Thus, the direction of the molecular dipole is parallel to the substrate; the electric field being perpendicular to the substrate has minimal impact on the molecular dipole. Therefore, molecular dipole of PhCN in electric field might influence the EL negligibly.

We have extended the above analysis in the revised manuscript (page 12).

Fig. R3-5 (a) PL spectrum of PhCN and EL spectra of devices I (b), II (c) and III (d).

Fig. R3-6 Schematic diagram of electron-hole recombination dynamics in PL and EL processes.

Comment 4: In line 31, the paper says that the commercial WLEDs composed of blue-emissive LEDs chips and yellow-emissive phosphors coating that “bring unstable emission color over time”, however, there is no data on device stability or device lifetime for phCN-based devices.

Supplementary data should be added to prove improved stability, otherwise it is difficult to demonstrate the potential of this material when both EQE and maximum luminance are much lower than those of commercial WLEDs.

Response: We thank the reviewer for this comment. As the reviewer's suggestion, we detected the device operational lifetime and spectral stability of the PhCN-based LEDs with different device structures (Fig. R3-7, also shown in Fig. S12 in Supplementary Information). Unfortunately, we obtained a short lifetime of the device, tens of minutes. It should be emphasized that the spectra of PhCN-based LEDs are very stable, and there is no obvious color shifting occurring during the operating process. The g-CN materials were generally studied as photocatalysts. As luminescent materials, g-CN was neglected and rarely investigated. However, the g-CN materials deserve more attention due to their high PLQY, ultra-low cost, eco-friendly nature and structural diversity. Of course, more efforts are required to in-depth understand and optimize the g-CN materials and their LEDs devices. We have added these results in the revised manuscript (page 12).

Fig. R3-7 (a) Device operational lifetime measured in constant current mode. Spectral stability of (b) device III and (c) device VI. Device IV: PEDOT:PSS/TFB/PVK (3.5 mg ml^{-1})/PhCN (5 nm)/B3PYMPM (40 nm)/LiF (1 nm)/Al (80 nm); device V: PEDOT:PSS/TFB/PVK (5 mg ml^{-1})/PhCN (5 nm)/1,3,5-tri(p-pyrid-3-yl-phenyl)benzene (TpPyPB, 40 nm)/LiF (1 nm)/Al (80 nm); device VI: PEDOT:PSS/TFB/PVK (8 mg ml^{-1})/PhCN (5 nm)/TpPyPB (40 nm)/LiF (1 nm)/Al (80 nm).

REVIEWER COMMENTS

Reviewer #1 (Remarks to the Author):

The authors have done additional work to address my questions, and this revised manuscript was improved substantially compared with the first version. In my opinion, this article at current level is publishable.

Reviewer #2 (Remarks to the Author):

The revision is checked and is acceptable. So it is suggested for publication in Nature Communications.

Reviewer #3 (Remarks to the Author):

The manuscript investigated modified CN materials for white light emitting device with high performances. The maximum external quantum efficiency of the device reached 1.2%, much better than those of precious reports. The material characteristics and emission device mechanism were systematically studied, of which results are of significant reference for the peer to deeply study the related materials. However, for this kind of materials, it was still uncertain whether they have a good development potential in the lighting field. Compared with the lighting demand, there are many lighting technologies, including the monomolecular white polymer materials and devices with much high performance compared to that of the current results. Specific questions are as follows for reference.

1. For lighting demand, it is better to provide power efficiency data.
2. How to precisely control the emission site of the modified CN materials, different emission sites will have a great impact on the emission spectrum, which expect to be discussed.
3. To discuss the stability of the lighting device and whether it has potential for possible applications.

Reviewer #1 (Remarks to the Author):

The authors have done additional work to address my questions, and this revised manuscript was improved substantially compared with the first version. In my opinion, this article at current level is publishable.

Response: We thank the reviewer for his/her valuable efforts in reviewing our manuscript and recommending the manuscript's publication.

Reviewer #2 (Remarks to the Author):

The revision is checked and is acceptable. So it is suggested for publication in Nature Communications.

Response: We thank the reviewer for his/her valuable efforts in reviewing our manuscript and recommending the manuscript's publication.

Reviewer #3 (Remarks to the Author):

The manuscript investigated modified CN materials for white light emitting device with high performances. The maximum external quantum efficiency of the device reached 1.2%, much better than those of precious reports. The material characteristics and emission device mechanism were systematically studied, of which results are of significant reference for the peer to deeply study the related materials. However, for this kind of materials, it was still uncertain whether they have a good development potential in the lighting field. Compared with the lighting demand, there are many lighting technologies, including the monomolecular white polymer materials and devices with much high performance compared to that of the current results. Specific questions are as follows for reference.

Response: We appreciate the reviewer for raising new comments to help us further improve the quality of the manuscript. In response, we have added new data in the manuscript.

Comment 1. For lighting demand, it is better to provide power efficiency data.

Response: According to the reviewer's suggestions, we have provided the power efficiency of the g-CN based LEDs (Fig. R3-1, also shown in Fig. 6f in the revised manuscript) and described in the revised manuscript (pages 10-12). On the one hand, g-CN material is a polymer semiconductor and has low electrical conductivity, which blocks the charges injection and results in high operating voltage and low power efficiency. Improving the conductivity of the material through molecular modification, doping, etc will be an effective way to further improve the device's performance. On the other hand, device structure influences the performance of LEDs significantly. The electron/hole transport layers with aligned energy levels and balanced charges injection are crucial to suppress exciton quenching and improve the device's efficiency. Overall, more effort is required to in-depth understand and optimize g-CN materials and their LEDs devices.

Fig. R3-1 EQE-voltage-power efficiency curves of devices I, II, IIc, and III, hollow circles represent EQE and solid circles represent power efficiency.

Comment 2. How to precisely control the emission site of the modified CN materials, different emission sites will have a great impact on the emission spectrum, which expect to be discussed.

Response: We thank the reviewer for the comment. The g-CN materials are mainly studied as photocatalysis, and their luminescence mechanism is rarely reported. Two kinds of methods, atomic modification and molecular modification were commonly applied to optimize the properties of the g-CN materials. (i) Atomic modification includes metal and nonmetal elements doping. The impurity atoms (e.g. B, P, O, S, Fe, Na) would change the electronic structure of the g-CN material, meanwhile positively charged or negatively charged centers could be formed in the g-CN network. These charged centers would attract electrons or holes, making electron-hole pairs more prone to recombine and luminesce near the centers. By controlling the content of doping elements, the number of charged centers and the emission of g-CN materials can be continuously controlled, which has been demonstrated in previous reports [Adv. Optical Mater. 2019, 7, 1900775; Adv. Optical Mater. 2016, 4, 2095; J. Mater. Chem. C 2016, 4, 6839]. Thus, the doped atoms could be considered as the emission sites. (ii) Molecular modification was used in our work. The g-CN material was modified by introducing phenyl group, which would enhance the degree of conjugation, promote electron delocalization and change the electronic structure of the g-CN. The electron and hole should diffuse in the whole conjugated system, and the emission site may be not limited to a certain atom.

Comment 3. To discuss the stability of the lighting device and whether it has potential for possible applications.

Response: We thank the reviewer for the comment. As preliminary attempt in the field of electroluminescence, g-CN materials will inevitably have many shortcomings. The stability of the g-CN based LEDs might be influenced by the g-CN materials and the device structure. (i) As polymer semiconductors, g-CN materials have high operating voltage due to their lower electrical conductivity. During device operation, the structure of g-CN materials may be changed and induce luminescence quenching. However, the spectral of the g-CN based LEDs is stable, indicating that the basic structure of the g-CN is kept and some defect sites form under the bias voltage, which would trap the electron-hole, induce non-radiative recombination and bring down the device's performance. (ii) The severe non-radiative recombination of electron-hole will heat the device. The thermal effect might influence the properties of the ETL/HTL material, electrode material, g-CN, and affect the interfacial interactions.

In our manuscript, we demonstrate the superior properties of the g-CN material, including excellent optical properties (high PLQY, tunable PL spectrum), structural diversity, and ultra-low-cost synthesis. The g-CN based LEDs exhibit broadband emission and excellent spectral stability, which have promising prospects in solid-state lighting and display. At present, g-CN has been neglected and underestimated as a photoelectric material, but it deserves more research and development.

REVIEWERS' COMMENTS

Reviewer #3 (Remarks to the Author):

The manuscript has been carefully revised and characterized the g-, Ph-CN chemical structure and film structures. It has carried out a study of EL performances through multiple device structures. Based on the principle of carrier injection balance, the performance of light-emitting devices has been gradually improved. A device with relatively high efficiency has been developed for this two-dimensional material, which has significant reference value for peers. However, no matter for lighting or display applications, it seems that there was a weak development prospect.

Reviewer #3 (Remarks to the Author):

The manuscript has been carefully revised and characterized the g-, Ph-CN chemical structure and film structures. It has carried out a study of EL performances through multiple device structures. Based on the principle of carrier injection balance, the performance of light-emitting devices has been gradually improved. A device with relatively high efficiency has been developed for this two-dimensional material, which has significant reference value for peers. However, no matter for lighting or display applications, it seems that there was a weak development prospect.

Response: We thank the reviewer very much for the positive comments. The g-CN materials have attracted attention because of their wide application in catalysis. Due to their advantages of simple synthesis, low cost, non-toxicity, structural diversity, and photoelectric effect, g-CN materials have been extended to a wider range of applications, including optoelectronic devices, cell imaging, biosensing, etc. In our manuscript, the g-CN materials are used as a kind of photoelectric material to prepare LEDs devices. Due to the unique optical properties (high PLQY, tunable PL wavelength) of the g-CN, the single-component emitting layer-based WLEDs exhibit broadband emission and excellent spectral stability, and relatively high performance. Our study demonstrates the great potential of g-CN as a luminescent material, which is expected to become another promising optoelectronic material after organic small molecules, polymers, quantum dots and perovskites, and has a promising prospect in lighting and display. However, the in-depth understanding and study of the photoelectric physical properties of g-CN materials is the prerequisite to promote its further application and development in the field of optoelectronics, which requires more researchers to invest more efforts. Thus, we believe that the g-CN is a promising material not only in catalysis but also in optoelectronics applications.